# Nanopore Direct RNA Sequencing Reveals the Short-Term Salt Stress Response in Maize Roots

**DOI:** 10.3390/plants13030405

**Published:** 2024-01-30

**Authors:** Shidong He, Hui Wang, Minghao Lv, Shun Li, Junhui Song, Rongxin Wang, Shaolong Jiang, Lijun Jiang, Shuxin Zhang, Xiang Li

**Affiliations:** National Key Laboratory of Wheat Improvement, College of Life Sciences, Shandong Agricultural University, Tai’an 271018, China; 2022110648@sdau.edu.cn (S.H.); 17863807657@163.com (H.W.); 2022110685@sdau.edu.cn (M.L.); 18054483312@139.com (S.L.); 15165388221@163.com (J.S.); lzinling@163.com (R.W.); sdau2020@126.com (S.J.); 15621348537@163.com (L.J.)

**Keywords:** salt stress, ONT DRS, transcriptome, DEGs, base modifications

## Abstract

Transcriptome analysis, relying on the cutting-edge sequencing of cDNA libraries, has become increasingly prevalent within functional genome studies. However, the dependence on cDNA in most RNA sequencing technologies restricts their ability to detect RNA base modifications. To address this limitation, the latest Oxford Nanopore Direct RNA Sequencing (ONT DRS) technology was employed to investigate the transcriptome of maize seedling roots under salt stress. This approach aimed to unveil both the RNA transcriptional profiles and alterations in base modifications. The analysis of the differential expression revealed a total of 1398 genes and 2223 transcripts that exhibited significant variation within the maize root system following brief exposure to salt stress. Enrichment analyses, such as the Gene Ontology (GO) and Kyoto Encyclopedia of Genes and Genomes (KEGG) pathway assessments, highlighted the predominant involvement of these differentially expressed genes (DEGs) in regulating ion homeostasis, nitrogen metabolism, amino acid metabolism, and the phytohormone signaling pathways. The protein–protein interaction (PPI) analysis showed the participation of various proteins related to glycolytic metabolism, nitrogen metabolism, amino acid metabolism, abscisic acid signaling, and the jasmonate signaling pathways. It was through this intricate molecular network that these proteins collaborated to safeguard root cells against salt-induced damage. Moreover, under salt stress conditions, the occurrence of variable shear events (AS) in RNA modifications diminished, the average length of poly(A) tails underwent a slight decrease, and the number of genes at the majority of the variable polyadenylation (APA) sites decreased. Additionally, the levels of N5-methylcytosine (m5C) and N6-methyladenosine (m6A) showed a reduction. These results provide insights into the mechanisms of early salt tolerance in maize.

## 1. Introduction

Soil salinization is widely acknowledged as a significant detriment to worldwide agricultural productivity and sustainability. This issue is exacerbated by the escalating effects of climate change and the scarcity of rainfall, resulting in a rapid expansion of degraded saline soils. As a consequence, global food security faces a formidable challenge. Approximately one-fifth of the Earth’s irrigated farmland, responsible for forty percent of the world’s food production, suffers from the adverse consequences of soil salinization. Particularly during the seedling stage, salt stress exerts a conspicuous impact on the plant’s root system, potentially inducing structural and functional damage that undermines its capacity to absorb water and essential nutrients [1,2]. In fact, salt stress imposes limitations on plant growth through both osmotic stress, occurring at an early stage, and the gradual onset of ionic cytotoxicity [3]. When plants are subjected to salt stress, glycosylinositol-phosphorylated ceramide sphingolipids on the cell membranes can detect and bind Na^+^ ions. Consequently, the Ca^2+^permeable channel modulator of calcium channels Activator 1, located on the plasma membrane, is activated, leading to an elevation in intracellular Ca^2+^ levels, thus initiating the salt overly sensitive (SOS) pathway [4]. This SOS pathway plays a vital role in bolstering the plant’s ability to withstand salt stress. Sitting downstream of this pathway is SOS1, a Na^+^/H^+^ anti-transporter protein situated within the plasma membrane. SOS1 facilitates the expulsion of excess Na^+^ from the cell and governs the long-range transportation of Na^+^ within plants [5]. The mitogen-activated protein kinase (MPK) signaling cascade also contributes to the plant’s response to salt stress, with the MPK6-mediated phosphorylation of SOS1 serving as a key mechanism [6]. Moreover, SOS3/S-type anion channel-associated binding protein 8 acts as a sensor for the augmented Ca^2+^ signaling within the cytoplasm. It recruits SOS2 to the plasma membrane and stimulates its activity. Subsequently, SOS2 phosphorylates SOS1, thereby enhancing the plant’s ability to tolerate salt stress by augmenting the Na^+^/H^+^ exchange activity [7].

ONT DRS was founded upon the voltage differential spanning the polymer membrane and the protein nanopore contained therein. When a solitary RNA molecule traverses the nanopore, a conspicuous disparity emerges amidst the two sides of the orifice, thereby facilitating the detection of signals. Due to the dissimilar electrically charged properties possessed by the four bases of AGCU, the discernment of the passing base can be accomplished via the disparities in the electrical signals, thus enabling sequencing [8]. Distinguished from previous next-generation sequencing methodologies, which grapple with restricted read lengths and bias originating from reverse transcription or amplification, ONT DRS directly accomplishes RNA sequencing, eluding prejudice and errors arising from the transposition of RNA to cDNA. It exerts a capacity to perceive the intact length of RNA and its modifications [8,9,10]. ONT DRS has triumphantly found application in transcript quantification at the transcriptomic level. Moreover, it has been employed to assess the length of poly(A) tails, as well as to investigate base modifications, such as N5-methylcytosine and N6-methyladenosine. These endeavors have demonstrated the potential of ONT DRS for unraveling the complexities of transcriptomes in studies encompassing human beings, *Caenorhabditis elegans*, *Arabidopsis thaliana*, *Phyllostachys edulis*, viruses, and rice transcriptomes [11,12,13,14,15,16,17].

Maize (*Zea mays* L.) is an extensively cultivated crop worldwide and serves as a vital source of sustenance for both human and animal consumption. With its cultivation spanning over approximately 200 million hectares, maize production exceeds a staggering 100 billion metric tons [18]. Gaining a deeper comprehension of the intricate maize transcriptome holds the potential to unearth the underlying mechanisms through which this resilient crop copes with the adverse effects of salinity. In this investigation, we undertook the arduous task of functionally categorizing the genes exhibiting DEGs through the complete sequencing of native mRNA molecules extracted from the intricate root system of maize. This process was conducted subsequent to subjecting the plants to three hours of exposure to saltwater treatment. The main objective was to unravel the multitude of metabolic pathways implicated in the plant’s response to salt stress while simultaneously conducting a comprehensive exploration of the modifications occurring in the base structure of RNA molecules. Successfully unraveling these intricate processes will enhance our understanding of the rapid response mechanisms deployed by plant roots to combat salt stress and establish a solid foundation for future endeavors aimed at improving the tolerance of crops to salinity.

## 2. Results

### 2.1. Reference Genome Matching Results

The RNA derived from maize root systems that underwent a 3-h exposure to salt stress was analyzed utilizing ONT DRS. Subsequently, the sequencing libraries control (CK) and stressed (ST) were created using purified polyadenylated RNA and loaded into the PromethION Flow Cell (R9.4.1), which was then sequenced using the PromethION sequencer. The CK and ST libraries collectively generated an impressive 13 gigabytes of data, producing over 10.5 megabytes of base pairs. The average read lengths for both the CK and ST libraries were 735.9 bp and 902.4 bp, respectively. These sequences were aligned to the reference genome of the maize B73 self-inbred line (Zm-B73-REFERENCE-NAM-5.0) using Minimap2, resulting in an alignment rate of over 94%, thereby indicating the acquisition of high-quality ONT DRS data (Appendix A). By comparing the transcript sets obtained from sequencing to those present in the reference annotations with the employment of SQANTI2, a total of 12,501 novel transcripts were detected in CK, while a total of 9291 novel transcripts were identified in ST (Appendix A), representing a reduction of 25.68%.

### 2.2. Novel Isoforms Function Annotation

The newly acquired transcripts from the two libraries underwent functional annotation in five databases, namely the GO, KEGG, Non-Redundant Protein Database (NR), Swissprot, and Clusters of Orthologous Groups (KOG). Specifically, 12,501 distinct transcripts in the CK library were annotated in GO (49.68%), KEGG (30.71%), NR (97.34%), Swissprot (70.99%), and KOG (54.10%), resulting in a total of 12,177 gene transcript annotations (Figure 1A). On the other hand, the annotation rates for the 9291 unique transcripts in the ST library across the databases were GO (45.32%), KEGG (31.49%), NR (96.58%), Swissprot (67.12%), and KOG (51.68%), resulting in a total of 8980 gene transcript annotations (Figure 1B).

Upon utilizing the NR database, the primary matches for both libraries, CK and ST, were *Zea mays* L. (Appendix A). The genes identified in the KEGG database were classified into cellular processes, environmental information processing, genetic information processing, metabolism, and organic systems. Each category was further divided into secondary functional groups. In the CK and ST datasets, the transcripts predominantly fell into the “global and overview maps” category, followed by “carbohydrate metabolism”, “folding, sorting, and sorting”, “folding, sorting, and degradation,” and “translation” (Appendix A).

The genes matched with the KOG database were allocated into 25 groups. The most abundant group in CK was “general function prediction only”, followed by “signal transduction mechanisms”, with the least abundant being “cell motility” (Appendix A). In contrast, the second highest group in ST was “post-translational modification, protein turnover, chaperones” (Appendix A), suggesting that short-term salt treatment may induce various post-translational modifications in proteins. By comparing the GO annotation results of the CK and ST novel isoforms, it became clear that the number of novel transcripts involved in the different functions in young maize roots following short-term salt stress generally exhibited a decreasing trend. Moreover, there were no “locomotion” novel transcripts annotated in ST, unlike in CK. However, novel transcripts related to the “immune system process” were annotated in ST compared to CK (Appendix A), indicating that salt stress triggered an immune response in the maize root cells.

### 2.3. DEGs and DETs Analysis

A total of 1398 differentially expressed genes (956 upregulated and 442 downregulated) and 2223 differentially expressed isoforms (1431 upregulated and 792 downregulated) were discovered between the CK and ST samples using the edgeR analysis tool (|log2 FoldChange| ≥ 1, FDR ≤ 0.005), as depicted in Figure 2A,B. The cluster analysis revealed distinct gene expression patterns between the CK and ST samples (Figure 2C). Notably, the upregulated DEGs with the highest expression levels included Zm00001eb112930, Zm00001eb255880, Zm00001eb256260, Zm00001eb201180, Zm00001eb403700, Zm00001eb397060, and Zm00001eb012040. Conversely, among the downregulated DEGs, Zm00001eb260610, Zm00001eb031840, Zm00001eb022890, Zm00001eb309770, and Zm00001eb164740 were the most significantly affected genes. The qRT-PCR results confirmed the accuracy and reproducibility of the ONT DRS detection results (Appendix A). Appendix A provides comprehensive information on all the differentially expressed genes.

### 2.4. Gene Ontology Analysis of DEGs

We conducted a GO enrichment analysis on the differentially expressed genes (DEGs) and successfully annotated 1128 genes, encompassing 41 GO categories, all possessing a Q value < 0.05, as revealed in the GO analysis of 1398 DEGs (Figure 3, Appendix A). In the context of salt stress, the DEGs exhibited a significant enrichment across various biological processes, cellular components, and molecular functional categories. Notably, the DEGs predominantly manifested in response to wounding, the regulation of the jasmonic acid-mediated signaling pathway, a response to auxin, the regulation of defense response, a response to abscisic acid, and a response to salt stress, indicating the intricate interplay of abiotic stress and phytohormonal signaling during the initial phase of salt stress. Remarkably, the enrichment of the genes associated with the integral component of the membrane, extracellular region, and plasma membrane in the cellular component category suggests a profound modification of cellular membrane constituents in response to salt stress. Additionally, the most enriched GO categories within the molecular function category encompassed metal ion binding, transcription factor activity, sequence-specific DNA binding, heme binding, and oxidoreductase activity, which further corroborated the occurrence of abiotic stress responses and subsequent alterations in the transcriptome upon salt treatment. The GO analysis further underscored the pivotal role of the genes related to oxidoreductase activity, osmoregulation, ion homeostasis regulation, and phytohormone signaling in modulating the transcriptional response of young maize roots to salt stress.

### 2.5. KEGG Pathway Analysis of DEGs

To enhance the analysis of the dynamic biological pathways in the DEGs under the influence of salt stress, we conducted a KEGG enrichment analysis for the DEGs (Figure 4A). Within this analysis of 1398 DEGs, 351 genes received annotations. The findings unveiled a significant enrichment of DEGs within vital KEGG pathways, such as the metabolic pathways; biosynthesis of secondary metabolites; plant hormone signal transduction; mitogen-activated protein kinase (MAPK) signaling pathway; alanine, aspartate, and glutamate metabolism; nitrogen metabolism; ABC transporters; and carotenoid biosynthesis. These pathways hold profound importance in terms of plant responses to both biotic and abiotic stresses. Specifically, in the early stages of salt stress, we observed a predominantly upregulated expression of the majority of genes involved in the regulation of phytohormone signaling, nitrogen metabolism, amino acid metabolism, and the MAPK signaling pathways (Figure 4B–E). The accuracy and reproducibility of the ONT DRS results were confirmed by a qRT-PCR of randomly selected genes with an increased or decreased transcript abundance in the ST treatment. Among the genes examined, the qRT-PCR expression data showed a similar trend to the ONT DRS data (Appendix A). 

### 2.6. PPI Analysis of DEGs

The String database (Confidence > 0.7) was employed for conducting the PPI analysis, which was then visually presented using Gephi (Figure 5). The outcomes exhibited that, within the PPI analysis, a total of 82 proteins actively participated in the response to salt stress stimulus. Among this group, certain notable proteins included the aldolase-type TIM barrel family protein, ferredoxin–nitrite reductase, glucose-6-phosphate 1-dehydrogenase, nitrate reductase, glutamate synthase, pyruvate kinase, and glutamate decarboxylase. Additionally, proteins such as the jasmonate ZIM domain-containing protein, AP2-EREBP transcription factor (AP2 domain-containing protein), BHL2 domain-containing protein, BHL2 domain-containing protein (containing protein), BHLH transcription factor (DNA binding protein), and TIFY domain/CCT motif transcription factor family protein were identified. The presence of these proteins within the PPI network indicated that a complex molecular network, consisting of nitrogen pathway-related proteins, glycolysis pathway-related proteins, amino acid metabolism pathway-related proteins, jasmonate signaling pathway-related proteins (ZIM domain, etc.), and abscisic acid signaling pathway-related proteins (PYL, etc.), actively safeguard maize root cells against salt-induced damage. By establishing intricate connections with other proteins, this molecular network plays a critical role in ensuring the protection of root cells from salt-induced damage. The following example showcases a segment of this intricate molecular network, which aids in fortifying the integrity of root cells against salt-induced damage.

### 2.7. AS Analysis and APA

To elucidate the impact of salt-induced stress on AS in maize roots, SUPPA2 was employed to identify AS events in each sample. AS refers to alternative splicing, which occurs during the gene expression process when one gene produces multiple different splice variants during transcription. Alternative splicing allows for the selection of different splice sites in pre-mRNA splicing, resulting in the production of different mRNA transcripts and, in turn, multiple proteins with different functions from the same gene. Within the transcriptome data, seven types of AS events were detected, namely skipping exon (SE), alternative 3′ splice site (A3), alternative 5′ splice site (A5), mutually exclusive exon (MX), retained intron (RI), alternative first exon (AF), and alternative last exon (AL). The analysis of the results revealed that the A3 events exhibited the highest proportion in both samples, followed by RI, A5, and SE (Figure 6A). Nonetheless, the occurrence of various AS events decreased following 3 h of salt stress treatment (Appendix A). By comparing the data of CK and ST, we determined which genes underwent AS under salt stress conditions, as well as the changes in their splicing variants expression. The list of genes with AS events can be found in Appendix A. This can help us further understand the response mechanism of maize to salt stress. Furthermore, the analysis of APA using the ONT DRS sequencing results and TAPIS indicated a decrease in the number of mRNA molecules at numerous APA sites after 3 h of salt stress treatment, with the exception of those possessing more than six APA sites, which increased in abundance (Figure 6B, Appendix A).

### 2.8. Poly(A) Tail Length Analysis and Methylation Analysis

The ONT DRS sequencing method allows for sequencing from the 3′ end to the 5′ end of mRNA molecules, and the inclusion of the poly(T) junctions preserves the integrity of the poly(A) information. Therefore, by analyzing the electrical signals from the sequencing, we were able to estimate the length of the poly(A) tail of RNA molecules. In our study, we utilized nanopolish polya for the poly(A) tail analysis to determine the poly(A) tail lengths of mRNAs from maize roots under ST conditions. Our findings indicated that the average poly(A) tail length of mRNAs from the maize roots under ST was slightly smaller than that of the CK (Figure 6C,D). Furthermore, we employed EpiNano and Tombo to analyze the ONT DRS sequencing data of the two libraries in order to investigate the m5C and m6A modifications. Our results revealed that there were 33,457 identified m5Cs in the CK group, whereas only 24,211 were identified in the ST treatment group. Additionally, there were 7004 transcripts in which m5C co-occurred between the two groups, suggesting significant changes in the cytosine modification sites after 3 h of salt treatment. (Figure 6E; Appendix A). Concerning the m6A modifications, 17,096 m6As were identified in the CK group, whereas 16,263 m6As were identified in the ST group, indicating a significant reduction in the number of m6A modification sites after the salt treatment (Appendix A). Additionally, of the 16,263 identified m6As, 8075 transcripts co-occurred m6A between the CK and ST groups, implying changes in the adenine modification sites after 3 h of salt treatment, although the quantitative changes were not significant (Figure 6F; Appendix A). Collectively, these findings suggest that the methylation status of m5C and m6A on mRNAs responds to salt stress treatment and subsequently regulates the transcription and translation of the related genes.

## 3. Discussion

The phenomenon of soil salinization is widely acknowledged as a significant challenge that poses threats to both global agricultural productivity and sustainability. A staggering 20 percent of the world’s irrigated farmland, responsible for producing 40 percent of the planet’s food, endures the detrimental effects of soil salinization [19]. Particularly during the seedling stage, the adverse effects of salt-induced stress on the intricate root system of plants becomes more evident, potentially causing damage to both the structure and function of the roots. Consequently, this weakens the plant’s capacity to absorb vital water and nutrients [1,2]. In order to deepen our comprehension of the intricate molecular mechanisms underlying the transient impact of salt stress on young maize roots, an analysis of the mRNA levels within these roots after being subjected to either a three-hour freshwater treatment or a three-hour salt treatment was conducted using ONT DRS technology. The examination revealed notable alterations in novel transcripts, DEGs, DETs, as well as modifications at the base level, all of which were dependent on the given treatment.

In addition to governing plant growth and development, phytohormones serve as mediators for an array of environmental stresses, such as salt stress, in normal circumstances, thereby regulating plant growth and adaptability [20]. Abscisic acid (ABA), jasmonic acid (JA), ethylene (ETH), and salicylic acid (SA) have been recognized as vital players in regulating plant responses to abiotic stresses [21]. In this investigation, we delved into the functionality of DEGs and discovered a profound enrichment and upregulation of the genes involved in phytohormone signaling (Figure 4B). The JA signaling pathway was triggered following salt stress in plants [22]. The F-box protein CORONATINE INSENSITIVE1, which forms the S-phase kinase-associated protein 1 (SKP1)/Cullin/F-box protein COI1 E3 ligase complex alongside SKP1 and CULLIN1, facilitated the degradation of the JASMONATE ZIM domain (JAZ) by the 26S proteasome. Within this study, ten ZIM domain genes were considerably upregulated in maize roots, thereby indicating the activation of the JA signaling pathway in maize under salt stress conditions (Figure 4B and Figure 5). Upon the removal of JAZ, the repressed transcription factors (e.g., MYC) activated the expression of the JA-responsive genes, ultimately resulting in the inhibition of primary root growth [23,24]. Ahmed et al. demonstrated that jasmonate signaling is a central component of biotic and abiotic stress responses, and exogenous jasmonate has the ability to rescue the growth of salt-sensitive plants [25]. ABA emerged as one of the most influential stress hormones in defense against salt stress [26,27]. During ABA signaling, ABA-dependent SUCROSE NONFERMENTING1-RELATED PROTEIN KINASE2 (SnRK2)-type protein kinases directly phosphorylated and activated AREBs/ABFs and ABI5 [28,29]. Studies indicated that in response to salt stress, ABA regulates transpiration in plants by controlling the opening and closing of the stomata [30], and ABA-activated SnRK2 is responsible for modulating osmotic homeostasis through the regulation of BRASSINOSTEROID INSENSITIVE 1-ASSOCIATED RECEPTOR KINASE 1 and Alpha-amylase 3-dependent starch catabolism, leading to the production of sugars and sugar-derived osmolytes [31]. The PYL4 A194T mutant has shed light on the significant role of PYL4/Protein Phosphatase 2C interaction in ABA signaling, displaying reduced stomatal conductance and enhanced water utilization in Arabidopsis thaliana [32]. Importantly, our investigation unveiled two differentially expressed PYL genes (including two PYL4 and one PYL5) that were significantly downregulated in the maize root system, thus emphasizing the crucial role of PYL in response to salt stress (Figure 4B and Figure 5). Two hours after exposing the maize to salt stress, the concentration of ABA in the leaves witnessed an increase, suggesting that ABA synthesis and accumulation represented an early response of maize to salt stress [33]. In the present study, a majority of auxin/indole-3-acetic acid genes were observed to be upregulated (Figure 4B), corroborating the findings of Sicilia et al. [34]. These phytohormones (JA, ABA, ETH, and SA) may serve as pivotal signals for maize in its endeavor to combat salt stress, triggering the expression of numerous downstream genes and enhancing maize’s tolerance to salt stress. The key DEGs that we identified within the phytohormone signaling pathway play indispensable roles under salt stress, thus highlighting the significance of phytohormone signaling as a noteworthy mechanism in maize’s response to salt stress.

In the absence of nitrogen fixation, nitrate serves as the primary source of nitrogen (N) for plant seedlings [35]. Plants assimilate nitrogen by converting it into nitrite through nitrate reductase (NR) and subsequently into NH^4+^ through nitrite reductase (NiR). Glutamine synthetase (GS) and glutamate synthase (GOGAT) further assimilate N into Gln and Glu, respectively, enabling the synthesis of various nitrogenous compounds [35]. NH^4+^ can also be linked to Glu by glutamate dehydrogenase (GDH). Our study revealed that the early stages of salt stress in maize roots prompted the upregulation of multiple synthetic genes involved in NR, NiR, GS, GDH, and GOGAT (Figure 4C), suggesting that the seedlings attempted to counteract the detrimental effects of salt stress by maintaining nitrogen metabolism. Asparagine, an important amino acid for the long-distance transport of organic nitrogen from source to sink, underwent catabolism facilitated by asparagine synthase (ASN) and asparaginase [36]. Moreover, plant cells can dynamically redistribute their limited nitrogen resources to establish new metabolic pathways [37]. Glutamate, serving as a crucial substrate for transaminases, facilitates the transfer of nitrogen to and from amino acids, a process vital for nitrogen utilization. Our analysis of ASN and transaminase genes showed that the majority of these genes were upregulated at the onset of salt stress.

In order to counterbalance the detrimental repercussions of stress induced by excessive salt, plants employ a mechanism wherein they produce compatible solutes, such as liberated amino acids, to mitigate the adverse effects of high salinity on osmotic equilibrium. Previous studies have revealed a positive correlation between heightened salt tolerance and the accumulation of the total liberated amino acids in certain crops [38,39]. In our investigation, a substantial amplification in the levels of liberated amino acids was observed in response to salt-induced stress (Figure 4D), suggesting that when faced with salinity, maize root cells considerably enhanced the synthesis and accumulation of liberated amino acids. This strategic response served to alleviate the cellular damage caused by salt stress by augmenting the concentration of intracellular solutes and facilitating water absorption and retention. The MAPK cascade served as a vital mediator of signal transduction, governing the pathways of hormonal signaling while acting as a guardian against the perils of stress-induced injuries [38]. Comprising a network of interconnected protein kinase classes, including the illustrious MAPK kinase, the esteemed MAPK kinase, and the venerable MAPK [40], this cascade propagated signals by means of phosphorylation, thereby enlivening the neighboring tiers of protein kinases. In a study by Xiong and Yang, the activation of rice-derived OsMAPK5 by ABA was evidenced, wherein its overexpression fostered a heightened resilience towards biotic and abiotic adversities [41]. In Arabidopsis thaliana, ABA signaling can evoke an augmentation of the MAPK kinase signaling cascade via enigmatic MAPK1 or MAPK2, fortifying the plant’s tolerance to the scourge of salt stress [42]. The revelatory expression of the genes responsible for overseeing a septet of protein phosphatases witnessed significant upregulation within this investigation (Figure 4E), thus corroborating the inducible activation of the MAPK cascade in response to the afflictions imposed by salt stress. With the unveiling of salt stress’s ability to incite the MAPK cascade reaction, a cascading sequence of phosphorylation and activation befell the kinases, thereby unleashing a plethora of downstream responses.

The importance of the post-transcriptional activities of mRNAs includes splicing, editing, capping, poly(A) trailing, and modification [12]. AS is a process that generates multiple mRNA transcripts from the same precursor mRNA in eukaryotes by using different splice sites. AS was first found in the calcitonin and immunoglobulin genes [43,44]. AS usually occurs in eukaryotes, which significantly increases the biodiversity of proteins that the genome can encode [45]. AS plays an important role in the post-transcriptional response of plants to abiotic stresses, and alterations in the ratio of splicing variants in response to stresses may play a role in plant adaptation to these stresses [46]. In this study, the number of various AS events decreased after 3 h of salt stress treatment, resulting in the highest percentage for A3 events in both samples, followed by RI, A5, and SE (Figure 6A; Appendix A). These changes may reflect the adaptive regulation of plants to salt stress, and the adjustment of the variability rate of splicing may play an important role in the adaptation of plants to abiotic stresses. APA is an extensively prevalent gene regulatory mechanism in organisms with cell nuclei, in which a single transcript can possess diverse polyadenylation sites. This allows for the generation of transcripts with distinct coding sequences or varying lengths of 3′ untranslated regions (3′UTRs). Consequently, a singular gene can produce multiple mRNA transcripts, thereby influencing the functionality, stability, localization, and translational efficiency of these transcripts. Furthermore, APA contributes to the development and response to non-living environmental stresses in plants. In our investigation, we observed alterations in mRNA APA patterns following brief exposure to high salinity. Specifically, there was an augmentation in the abundance of mRNAs featuring multiple APA sites, suggesting an increased diversity that could potentially impact gene expression and protein function. Conversely, a decrease was observed in the number of other mRNAs exhibiting fewer APA sites, indicating a propensity towards specific transcripts. These findings could be linked to the regulation and function of specific genes (Figure 6B) [47,48,49]. The poly(A) tail is a constantly evolving attribute of eukaryotic mRNAs, with variations in its extent playing a pivotal role in governing gene expression through the modulation of nuclear export, mRNA stability, and translation dynamics [50]. The mean length of the poly(A) tail in the maize root systems subjected to salt ST was marginally smaller compared to CK (Figure 6C,D). This reduction in the poly(A) tail length possibly stemmed from the impact of salt stress on intracellular mRNA in the maize root system, ultimately leading to a slight decrease in the poly(A) tail extension. This phenomenon could potentially signify one of the cellular adaptations triggered by salt stress, aimed at fine-tuning the gene expression to suit the demands of the fluctuating environmental conditions. However, further exploration and investigation are needed to fully elucidate the specific regulatory mechanisms and contributing factors involved in this process.

ONT DRS, with its ability to directly sequence RNA without the need for reverse transcription and amplification processes, offers a more precise detection of the base modifications. mRNA molecules play vital regulatory roles at the post-transcriptional level, with m6A and m5C having been identified as key players in development and stress responses [51]. DNA methylation, encompassing m5C and m6A, is found in the genomic DNA of prokaryotes, archaea, and eukaryotes, thus holding significant importance as an epigenetic marker [52,53]. Studies on the m5C methyltransferase osnsnu2 in rice demonstrated that the osnsnu2 mutant exhibits a heat-insensitive phenotype, and heat stress further enhances the m5C modification of mRNAs involved in photosynthesis and detoxification processes [53]. m6A modifications play a critical regulatory role in plant yield development, nutrient growth, and stress adaptation. For instance, under salt stress conditions, m6A modification promoted the expression of salt-tolerant genes, such as SbIAR4 (SORBI_3010G101300) and SbNRT1.5 (SORBI_3004G276200), thereby enhancing the salt tolerance of sweet sorghum [54]. In Arabidopsis thaliana, m6A modifications also contributed significantly to salt stress tolerance. Mutants with substantially reduced m6A levels exhibited a salt-sensitive phenotype, and VIR-mediated m6A methylation negatively regulated reactive oxygen species homeostasis by influencing the elongation of the 3ʹ-UTR associated with alternative polyadenylation, as well as the mRNA stability of several negative regulators of salt stress, including Arabidopsis thaliana Activation Factor 1, GIGANTEA, and Glutathione S-Transferase U17 [55]. Under salt stress conditions, the degree of DNA methylation in certain plants can be significantly diminished, likely due to the impact of salt stress on the activity or stability of DNA methylation enzymes [55,56]. According to our research findings, we observed a significant decrease in the quantity of m5C modification and a slight decrease in the quantity of m6A modification after short-term salt stress (Figure 6E,F). This suggested that salt stress may affect the combined activity of RNA methyltransferases and demethylases in maize roots, leading to a decrease in m5C and m6A modifications, which was consistent with the previous research findings. In addition, we propose a new possibility that the changes in the m5C and m6A modification levels after short-term salt stress may contribute to the regulation of transient stress responses in maize plants. These findings highlight the regulatory effect of salt stress on methylation modifications in plants. Further investigations will shed light on the regulatory network and associated biological functions of methylation modifications under salt stress, ultimately deepening our understanding of the mechanism by which plants respond to salt stress.

Through transcriptome level analysis, the intricate processes governing gene expression in juvenile maize roots subjected to high concentrations of salt stress can be elucidated. Moreover, the signaling pathways that potentially participate in these processes can be unveiled. Such comprehensive understanding empowers us to gain deeper insights into the intricate mechanisms by which plants respond to salt stress. Furthermore, this knowledge serves as a valuable reference for enhancing crop diversity and breeding cultivars resilient to saline soils.

## 4. Materials and Methods

### 4.1. Plant Material and Sampling

B73 maize inbred seeds were hydroponically cultured in the artificial climate chamber at the National Key Laboratory of Wheat Breeding, Shandong Agricultural University. The hydroponic culture was conducted under a temperature regime of 28 °C/25 °C and a photoperiod of 16 h light/8 h dark. The maize plants were grown until the two-leaf stage. The root systems of the maize seedlings were then treated in water and a 150 mM NaCl solution, respectively, for 3 h [57,58]. In this study, exposing the plants to a 150 mM NaCl solution for 3 h was defined as short-term salt stress, which was used to observe early plant responses and rapid physiological changes to salt stress. Since the root apex of plants is an important growth point and active metabolic area, it is very sensitive to environmental stimuli. Therefore, the maize root tips were collected and frozen in liquid nitrogen and stored at −80 °C for further use.

### 4.2. RNA Extraction and Library Preparation

Total RNA was extracted using TRIzol (Invitrogen, Carlsbad, CA, USA) by following the user manual. The RNA degradation and contamination of each sample was monitored on a 1.5% agarose gel. Using the NanoDrop One UV–Vis spectrophotometer (Thermo Fisher Scientific, Waltham, MA, USA), the RNA purity was assessed, followed by precise quantification using the Qubit^®^ 3.0 fluorometer (Invitrogen, USA). The enrichment of mRNA (polyA+ RNA) was performed using the NEBNext Poly(A) mRNA Magnetic Separation Module (E7490), and the enriched mRNA was subjected to analysis using the Direct RNA Sequencing Kit (Oxford Nanopore Technologies, Oxford, UK, SQK-RNA002). The reverse transcription adapters (RTAs) were ligated to the 3′ end of the enriched mRNA molecules using the Direct RNA Sequencing Kit (Oxford Nanopore Technologies, SQK-RNA002). The mRNA molecules were reverse transcribed, synthesizing their complementary strands, and the RTAs were attached to the sequencing adapters located at the RTA terminus, resulting in the final sequencing library. Following the library construction, the precise quantification of the pre-sequencing library was conducted using the QubitTM dsDNA HS Assay Kit (Invitrogen, LOT 2133187). After the library quality control, the prepared sequencing library was loaded onto the PromethION Flow Cell (R9.4.1) chip and subjected to sequencing on the PromethION sequencing instrument.

### 4.3. Basecalling and Mapping

The downstream data of the sequencing were saved in the FAST5 format, and these readings were calibrated using the default RNA parameters in the GUPPY. The original Fastq reads were filtered using NanoFilt (v2.5.0) [59]. The calibrated reads were then compared to a reference genome using Minimap2. Subsequently, Stringtie (v2.1.4) was utilized for redundant clustering and sequence correction of the reads based on the reference genome [60,61]. Finally, Salmon (v0.14.1) was employed for transcript and gene quantification [62]. The obtained non-redundant transcriptome set from sequencing was compared to the transcriptome set in the reference annotation using SQANTI2 (v3.8), which generated a new enhanced transcriptome reference file for maize genomes [63].

### 4.4. Differentially Expressed Genes (DEGs) and Transcripts (DETs) Analysis

Using Minimap2, the reads from each sample were compared against the reference transcriptome sequence, with the comparison parameters set as -ax splice -uf --junc-bed [60]. Then, Salmon (v0.14.1) was used for the quantification of the transcripts and genes, with the parameters set as -noErrorModel -l U [62]. The DEGs and DETs, defined as |log2FoldChange| ≥ 1 and FDR ≤ 0.005, were computed using the R package edgeR (v3.0.8) [64].

### 4.5. Functional Enrichment and PPI Analysis

The enrichment analysis of DEGs was conducted using DAVID (https://david.ncifcrf.gov/tools.jsp (accessed on 26 March 2023)) [65]. STRING (https://cn.string-db.org/ (accessed on 25 April 2023)) was utilized for the PPI networks of the DEGs [66]. The resulting PPI networks were imported into Cytoscape (v3.9.1), where the nodal degrees of freedom were calculated using the CytoNCA plugin [67]. The visualization of the networks was performed using Gephi (v0.9.7), ensuring a confidence level exceeding 0.7.

### 4.6. AS and APA Analysis

The identification of variable shear events for each sample was carried out using SUPPA2 (v2.3) with the parameters adjusted to -e SE SS MX RI FL [68]. Subsequently, the count of each variable shear event and its distribution were determined. For the analysis of variable polyadenylation, TAPIS (v1.0) was employed with the parameters set to -s 2 [69].

### 4.7. Poly(A) Length Estimation and Methylation Analysis

Nanopolish polya (v0.11.2) was utilized to calculate the length of the poly(A) tail for each read [14]. Subsequently, Tombo (v1.5.1) was employed to analyze m5C methylation for each sample, with the parameter—alternate-bases 5mC—RNA. Additionally, EpiNano (v1.1) was employed to predict the m6A sites, while the resulting predictions underwent filtration [70].

### 4.8. Quantitative Real-Time PCR (qRT-PCR) Analysis

The RNA samples employed for the ONT DRS experiments underwent a real-time PCR analysis to verify the dependability and reproducibility of the findings. In order to eliminate any contamination from genomic DNA, the total RNA was treated with DNase I (RNase Free) (Takara, Dalian, China). Subsequently, the cDNA was synthesized through a reverse transcription reaction utilizing random primers (Promega, Madison, WI, USA). The resulting cDNA was then subjected to analysis on a 7500 Real-Time PCR System (Applied Biosystems, Waltham, MA, USA), employing the Power SYBR Green PCR Master Mix (Applied Biosystems, Foster City, CA, USA) in accordance with the manufacturer’s guidelines. All three biological replicates were assessed accordingly. Gene-specific primers were designed using the Primer 3 software [71], and the Primer designs are shown in Appendix A. The relative values of the expression levels were calculated using the 2^−ΔΔCt^ method [72].

## 5. Conclusions

The salt stress conditions had an impact on the overall physiological metabolism of the maize root system, eliciting the induction of relevant genes. An ONT DRS analysis identified a total of 1398 DEGs and 2223 DETs in response to salt stress for a duration of 3 h. Enrichment analyses based on the GO and KEGG pathways revealed the involvement of these DEGs in the regulation of ion homeostasis, osmotic regulation, metabolic pathways, and phytohormone signaling. Through PPI analyses, it was observed that a complex network of molecules, including proteins related to the glycolytic metabolic pathways, nitrogen metabolic pathways, amino acid metabolism pathways, abscisic acid signaling pathways, and jasmonic acid signaling pathways, worked together to safeguard the root cells against salt-induced damage. Furthermore, under salt stress conditions, the modifications in the base sequences of maize root mRNAs tended to be simplified, with a decrease in the occurrence of AS events in RNA modifications, a slight reduction in the average length of poly(A) tails, and a decrease in the number of genes at the majority of the APA sites, as well as a reduction in the number of m5C and m6A methylations. These findings provide valuable insights into the functionality of the genes associated with salt tolerance.

## Figures and Tables

**Figure 1 plants-13-00405-f001:**
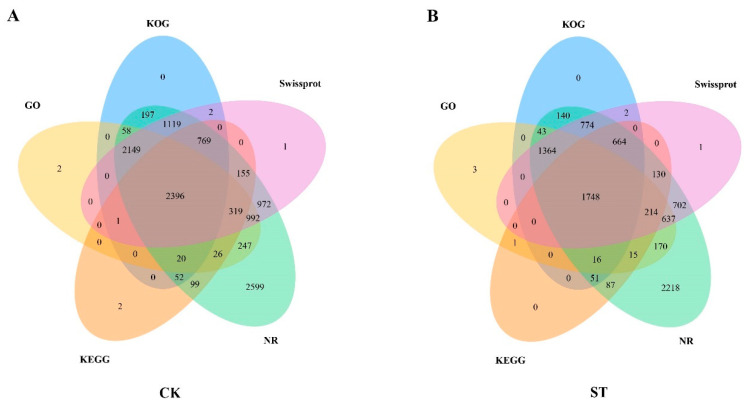
Venn diagrams showcasing the novel isoforms of CK and ST across five significant databases. (**A**,**B**) The GO, KOG, Swissport, NR, and KEGG databases are symbolized by yellow, blue, purple, green, and orange ovals, respectively. The numerical values indicate the count of newly discovered transcripts annotated in the various databases.

**Figure 2 plants-13-00405-f002:**
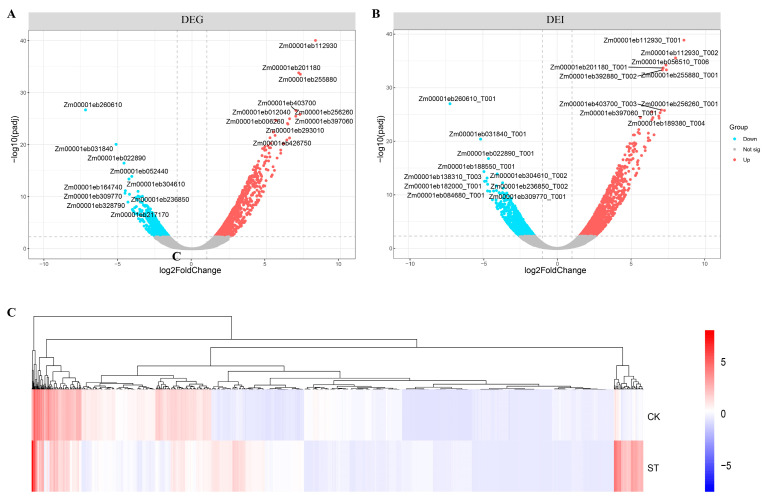
Volcano plots of the DEGs and DETs of CK and ST and the cluster analysis of the DEGs. (**A**) The red color indicates the upregulated genes and the light blue color indicates the downregulated genes. (**B**) The red color indicates the upregulated transcripts and the light blue color indicates the downregulated transcripts. (**C**) The rows indicate the samples; the columns indicate the different genes. The colors indicate the expression levels of the genes in the samples as determined by RPG10K.

**Figure 3 plants-13-00405-f003:**
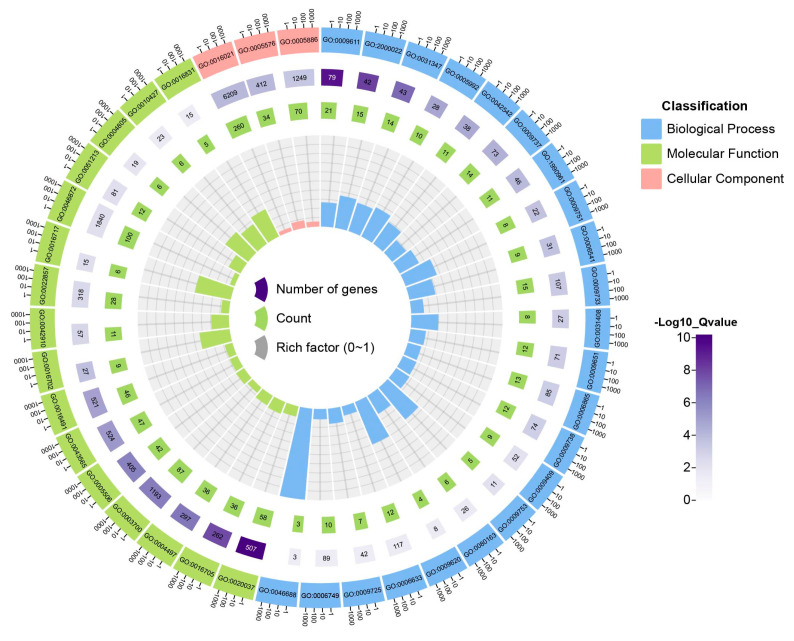
GO enrichment analysis for DEGs. The initial circle exhibits the identification data of GO, where the distinct hues represent various Gene Ontologies. Subsequently, the second circle displays the quantity and Q values of the GO elements within the background gene set, with lengthier bars and darker shades suggesting a larger number of genes and a lower Q value, respectively. The third circle showcases the count of DEGs annotated to GO items, with lengthier bars symbolizing a greater number of genes. Lastly, the fourth circle reveals the ratio between the number of DEGs annotated to a particular GO item and the number of genes annotated to said GO item within the background gene set. Furthermore, gridlines divide the background into increments of 0.1.

**Figure 4 plants-13-00405-f004:**
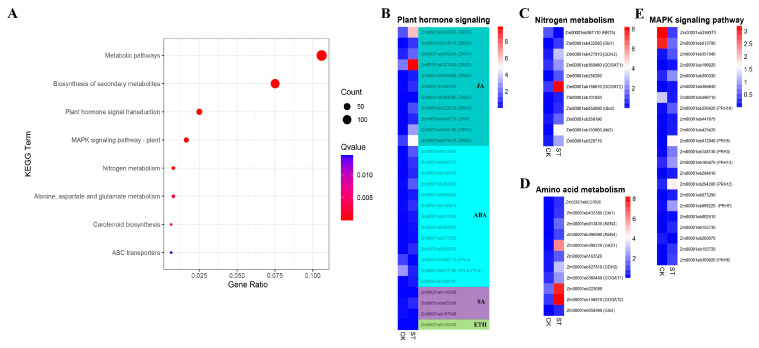
KEGG pathway enrichment analysis and heat map of the expression of genes related to plant hormone signaling, nitrogen metabolism, amino acid metabolism, and the MAPK signaling pathways (**A**) The y-axis indicates the pathway of the KEGG, and the gene ratio on the x-axis indicates the ratio of the number of DEGs annotated to this pathway to the number of genes in the background genes annotated to this pathway. (**B**–**E**) The columns indicate the samples and the rows indicate the different genes. The color indicates the expression level of the gene in the sample as determined by RPG10K.

**Figure 5 plants-13-00405-f005:**
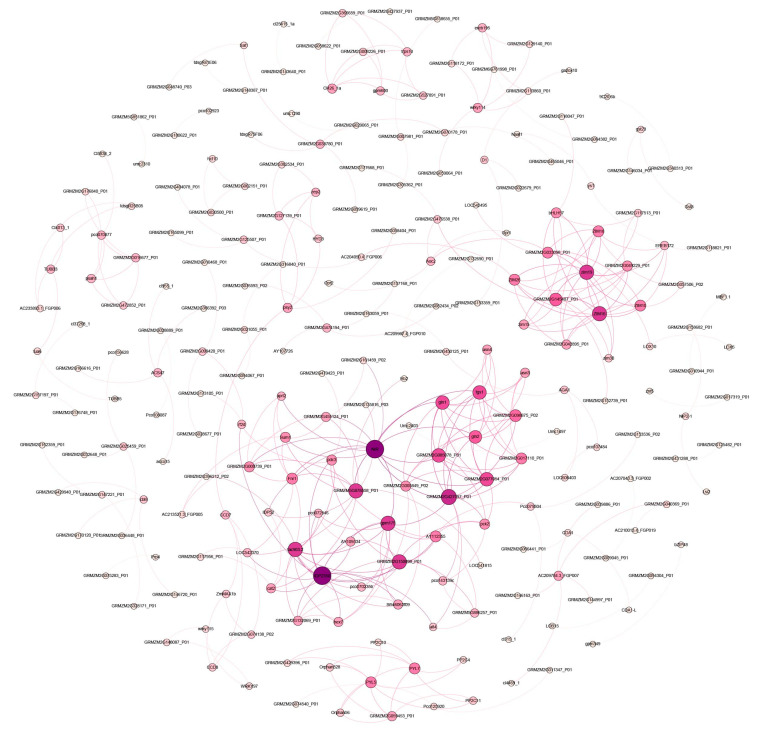
PPI network of the DEGs. The size of the node indicates the size of the degrees of freedom, calculated using the Cytoscape (v3.9.1) plugin CytoNCA.

**Figure 6 plants-13-00405-f006:**
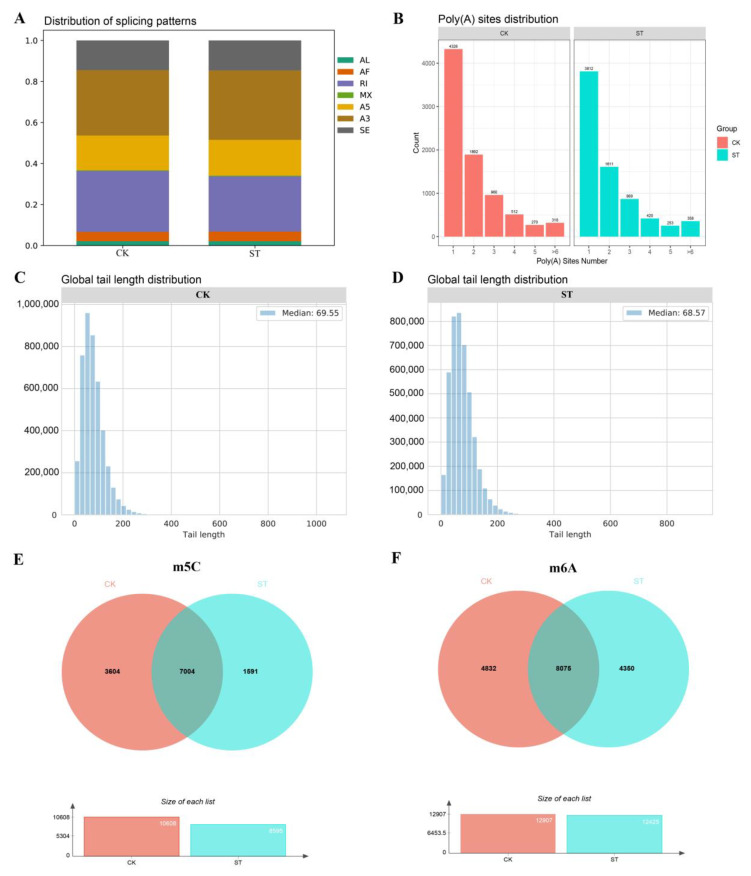
Distribution of each variable shear event, a plot of the number of genes with different numbers of APA sites, and a plot of the overall poly(A) tail length within the sample. (**A**) The diverse colors correspond to distinct AS events, while the size of the graph signifies the proportion of a specific AS event relative to the total AS events. (**B**) The x-axis denotes the number of distinct APA sites, while the y-axis represents the amount of genes displaying varying numbers of APA sites. (**C**,**D**) The x-axis indicates the RNA poly(A) tail length, while the y-axis indicates the amount of RNA possessing different poly(A) tail lengths. (**E**) The orange color indicates the number of transcripts where m5C was present in CK, the light blue color represents the number of transcripts where m5C was observed in ST, and the overlapping region indicates the number of transcripts where m5C co-occurred in both groups. (**F**) The orange color signifies the number of transcripts in CK featuring m6A, the light blue color corresponds to the number of transcripts in ST with m6A, and the intersection denotes the number of transcripts where m6A was simultaneously present in both groups.

## Data Availability

The sequencing datasets generated by ONT DRS are available in the NCBI Sequence Read Archive (SRA) BioProject: PRJNA1012554.

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
