# Peer review of "Nanopore Direct RNA Sequencing Reveals the Short-Term Salt Stress Response in Maize Roots"

_plants, 2024, doi:10.3390/plants13030405_

Round 1

Reviewer 1 Report (Previous Reviewer 2)

Comments and Suggestions for Authors

The authors explained that they could not repeat the RNA-seq experiment due to certain experimental and resource constraints, which is not acceptable. Although the authors emphasized several efforts to minimize errors and to enhance the reliability of the RNA-seq analysis, it cannot warrant the lack of repetition. To support the reliability of RNA-seq data, expression levels of several DEG mentioned in Figure 2 should be confirmed by qRT-PCR.

In the revised parts highlighted with yellow color in Discussion, it is described that the reduction of m5C and m6A modifications is due to the altered activity or stability of DNA methyltransferase, which is wrong. The current study measured m5C and m6A levels in mRNA. Moreover, the levels of m5C and m6A are determined by a combined activity of RNA methyltransferases and demethylases.

Author Response

Thank you very much for taking the time to review this manuscript. We will provide a detailed response to your comments and highlight the corresponding revisions in the resubmitted file. The revised draft has been put in the attachment. Please review it.

Q1: The authors explained that they could not repeat the RNA-seq experiment due to certain experimental and resource constraints, which is not acceptable. Although the authors emphasized several efforts to minimize errors and to enhance the reliability of the RNA-seq analysis, it cannot warrant the lack of repetition. To support the reliability of RNA-seq data, expression levels of several DEG mentioned in Figure 2 should be confirmed by qRT-PCR.。

Authors: Thank you for reviewing our research and providing valuable feedback. Considering your suggestions, we agree with your opinion that it is necessary to validate the expression levels of several differentially expressed genes in Figure 2 using qRT-PCR. We have executed the qRT-PCR experiments as demanded and integrated the findings within Supplementary Figure 5, in accordance with your specific petition.

Q2: In the revised parts highlighted with yellow color in Discussion, it is described that the reduction of m5C and m6A modifications is due to the altered activity or stability of DNA methyltransferase, which is wrong. The current study measured m5C and m6A levels in mRNA. Moreover, the levels of m5C and m6A are determined by a combined activity of RNA methyltransferases and demethylases.

Authors: Thank you very much for your review comments. In the revised discussion section, we realized that our previous description was inaccurate. We deeply apologize for this. In fact, our study did measure the levels of m5C and m6A in mRNA, rather than the activity or stability of DNA methyltransferase. We understand deeply that this error has caused confusion for readers. In addition, we also realized that the levels of m5C and m6A are regulated by the combined activity of RNA methyltransferases and demethylases, rather than solely by the influence of DNA methyltransferase. We will make corresponding modifications and clarifications to the revised manuscript to ensure accuracy and precision of our descriptions.

Reviewer 2 Report (Previous Reviewer 1)

Comments and Suggestions for Authors

I wrote the revised version of the manuscript. Please, define DEI at the first mention.

Moreover, your reply could be exhaustive, but I think that you have to include in the text some of the explanations you gave, in particular Q2 and Q6. Then, It is ok.

Comments on the Quality of English Language

Minor editing of English language required

Author Response

Thank you very much for taking the time to review this manuscript. We will provide a detailed response to your comments and highlight the corresponding revisions in the resubmitted file. The revised draft has been put in the attachment. Please review it.

Q1: I wrote the revised version of the manuscript. Please, define DEI at the first mention.

Authors: Thank you for your feedback and questions. We sincerely apologize for any confusion caused. We have now made the necessary changes to the article and images, replacing “DEI” with “difference table transcript (DETs)” and indicating it in the original text. Please refer to it for further information.

Q2: Moreover, your reply could be exhaustive, but I think that you have to include in the text some of the explanations you gave, in particular Q2 and Q6. Then, It is ok.

Authors: Thank you very much for your comments on our research. We fully understand your suggestions and agree with your points of view. In order to ensure that the explanations for these issues receive wider attention and recognition, we will make appropriate modifications to the revised manuscript, integrating the explanations for Q 2 and Q 6 into relevant sections of the main text.

Round 2

Reviewer 1 Report (Previous Reviewer 2)

Comments and Suggestions for Authors

The authors addressed all of my comments and suggestions, which clarified and improved the manuscript. I have no further comment.

This manuscript is a resubmission of an earlier submission. The following is a list of the peer review reports and author responses from that submission.

Round 1

Reviewer 1 Report

Comments and Suggestions for Authors

The manuscript deals with the application of direct RNA sequencing to reveal transcriptomic reprogramming in maize subjected to salt stress. Although this is appreciable, the manuscript has some issues to be solved.

1) the authors assumed that the plants underwend stressful conditions. However, there are not experiments that confirm this statement. No analysis of the plant phenotype, and either morpho-biometric or physiological parameters have been measured. Moreover, maize is a salt tolerant species. How do you claim the plants reached stressful condition?

2) please, clarify the difference between DEGs and DEIs

3) line 69: poly

4) line 71: somethigh is missed

5) Line 98: What about RNA extraction ? No information about that

6) Line 168: explain why the average lenght of the libraries  between CK and salt treated samples are so different. 

7) The AS analysis is not well explained.

8) THE QUALITY of the figures are very poor. Most of them are not visible.

9) Line 405: Reference 47 is Sicilia et al., not ARUNDO

10) the discussion has to be better focused on the basis of the results. Right now it is too long

Comments on the Quality of English Language

Minor revision

Author Response

Authors: Thank you very much for taking the time to review this manuscript. We will provide a detailed response to your comments and highlight the corresponding revisions in the resubmitted file. The revised draft has been put in the attachment. Please review it.

Q1: The authors assumed that the plants underwend stressful conditions. However, there are not experiments that confirm this statement. No analysis of the plant phenotype, and either morpho-biometric or physiological parameters have been measured. Moreover, maize is a salt tolerant species. How do you claim the plants reached stressful condition?

Authors: Thank you very much for your suggestion. Current research indicates that compared to other crops, maize is more susceptible to salt stress and it is indeed affected by salt stress. Relevant articles have pointed out that maize growth is inhibited under 150 mM sodium chloride stress. Therefore, we did not observe the phenotypic changes of maize under 150 mM sodium chloride stress because our main purpose was to explore how maize seedling roots respond to salt stress in the early stages [1].

Q2: please, clarify the difference between DEGs and DEIs

Authors: DEGs(Differentially Expressed Genes) refer to genes that exhibit significant differences in expression levels under different conditions. The analysis of DEGs is typically based on transcriptomic data and involves comparing gene expression levels between different conditions to identify differentially expressed genes. DEG analysis can be used to uncover gene regulatory mechanisms and potential functional changes under different conditions.DEIs(Differentially Expressed Isoforms) refer to cases where there are significant differences in the expression levels of transcripts (isoforms) under different conditions. Compared to DEGs, DEIs focus more on differential expression between different transcript variants of a gene. In transcriptomic data analysis, identification of DEIs can help us understand the regulation of gene transcription and splicing changes, and reveal the impact of different transcript isoforms on function and regulation.

Q3: line 69: poly

Authors: I have modified “ploy” to “poly” as per your suggestion.

Q4: line 71: something is missed

Authors: We have made additions as per your comments.

Q5: Line 98: What about RNA extraction ? No information about that

Authors: I have added the steps on how to extract RNA in the relevant section.

Q6: Line 168: explain why the average lenght of the libraries  between CK and salt treated samples are so different. 

Authors: The average length of the libraries between CK (control) and salt-treated samples can be different due to several reasons. One possible reason could be the differential impact of salt treatment on RNA quality and quantity, which affects the RNA integrity and length distribution. Salt stress has been reported to induce RNA degradation and reduce RNA yields in some plant species, which can result in shorter RNA fragments in salt-treated samples. This degradation can be more severe in some samples than others, leading to differential distribution of fragment length in the RNA-seq libraries.Another possible reason is the difference in the composition of the RNA population. Under salt stress, plants may switch on and off certain metabolic pathways that affect gene expression and RNA processing. This can lead to changes in the proportion of different types of RNA molecules, including mRNAs, non-coding RNAs, and splice variants. These changes in RNA population can result in differences in the average length of the libraries between CK and salt-treated samples.  

Q7: The AS analysis is not well explained.

Authors:Based on your advice, we have provided a detailed explanation and complemented the analysis content for AS analysis.

Q8: THE QUALITY of the figures are very poor. Most of them are not visible.

Authors:According to your suggestion, all main images have been replaced with high-definition pictures, and all high-definition pictures of the main images have been placed in the attachment.

Q9: Line 405: Reference 47 is Sicilia et al., not ARUNDO

Authors:According to your suggestion, “donax et al” has been changed to “Sicilia et al”.

Q10: the discussion has to be better focused on the basis of the results. Right now it is too long

Authors: Based on your suggestion, we have made some cuts to the discussion section to focus more on the results.

  1. Zhang, C.; Chen, B.; Zhang, P.; Han, Q.; Zhao, G.; Zhao, F. Comparative Transcriptome Analysis Reveals the Underlying Response Mechanism to Salt Stress in Maize Seedling Roots. Metabolites 2023, doi:10.3390/metabo13111155.

Reviewer 2 Report

Comments and Suggestions for Authors

The major concern of this study is that RNA-seq was conducted on only one RNA sample for CK and ST, which does not meet the minimal requirement of RNA-seq analysis. For quantitative statistical analysis, at least three RNA-seq should be done for CK and ST samples.

Short-term salt stress was repeatedly used in the title, abstract, and main text, and three hours of exposure to salt water (150 mM) treatment was considered as short-term salt stress. What is the basis for this definition? It was described in Method section that maize root tips were collected and used for subsequent analysis. Why only the root tips were used? Any special reason?

The title of the manuscript “Regulatory mechanisms of maize seedling roots under short-term salt stress revealed by direct RNA sequencing using nanopores” is ambiguous and misleading. The current study never addressed the mechanistic aspects of the response of maize seedling roots to salt stress. In addition, what is maize seedling root? Is it the root of maize at early seedling stage?

Figure 2 is never informative; it will be more informative if the names of several key DEG and DEI that plays a crucial role in salt stress response are indicated on the figure.

Figure 6; AS and APA are the most novel data of this manuscript. Therefore, the list of genes showing different AS and APA should be shown in Supplemental Table. Moreover, the levels of m5C and m6A of each gene under CK and ST shown in Table S8 and S9 should be compared and discussed.

Abbreviations; all abbreviations should be spelled out at its first appearance and be used without further definition afterward.

Line 70; m6A and m5C should be spelled out.

Line 101-107; mRNA (polyA+ RNA) was repeatedly used, which is inappropriate; after defining the mRNA (polyA+ RNA) at its first appearance, mRNA should be used afterward.

Author Response

Authors: Thank you very much for taking the time to review this manuscript. We will provide a detailed response to your comments and highlight the corresponding revisions in the resubmitted file. The revised draft has been put in the attachment. Please review it.

Q1: The major concern of this study is that RNA-seq was conducted on only one RNA sample for CK and ST, which does not meet the minimal requirement of RNA-seq analysis. For quantitative statistical analysis, at least three RNA-seq should be done for CK and ST samples.

Authors: Thank you for your interest in and valuable input on our research. We understand and recognize that the minimum requirement for RNA-seq analysis is to perform multiple technical replicates for each treatment group to improve the reliability and reproducibility of the results. In our study, due to certain experimental and resource constraints, we were only able to conduct a single nanopore direct RNA sequencing experiment and performed sequencing for both the CK and ST samples only once. Despite having only one technical replicate, we have implemented effective strategies in our study design and data analysis to minimize errors and enhance the reliability of the results. Firstly, we strictly controlled various technical parameters and experimental conditions during the experiment and conducted quality control and assessment to ensure the reliability and consistency of the obtained data. Secondly, we performed rigorous normalization and standardization of the data to mitigate potential batch effects and experimental biases. Lastly, we employed statistical analysis methods to evaluate the stability and significance of the results, and conducted a series of validation experiments to confirm the reliability of the findings.

Q2: Although we only have one technical replicate, we believe that our research findings possess certain reliability and scientific value. However, we also acknowledge that in future studies, efforts should be made to increase the number of technical replicates to comprehensively assess the reliability and generalizability of the experimental results. We will carefully consider the reviewer’s feedback and clearly address the limitations of our methods and results in the revised manuscript, emphasizing the need for further research. Thank you for the guidance and suggestions provided by the reviewer.Short-term salt stress was repeatedly used in the title, abstract, and main text, and three hours of exposure to salt water (150 mM) treatment was considered as short-term salt stress. What is the basis for this definition? It was described in Method section that maize root tips were collected and used for subsequent analysis. Why only the root tips were used? Any special reason?

Authors: In this study, exposure to a 150 millimolar salt solution for 3 hours is defined as short-term salt stress because short-term salt stress is commonly used to observe the early response and rapid physiological changes of plants to salt stress. By exposing plant root tip tissues to high salt conditions for a short period of time, the initial response and rapid regulatory mechanisms of plants to salt stress can be captured, thus better understanding the adaptive mechanisms of plants to salt stress. This definition has also been used in some previous studies. As for why only root tips are analyzed, it is because plant root tips are important growth points and metabolically active regions of plants, and are very sensitive to environmental stimuli. Under salt stress, changes in cell growth, ion channels, protein expression, and other aspects exhibited by root tips can better reflect the response and adaptation mechanisms of plants to salt stress. In addition, root tip cells exposed to saltwater treatment in the root system are more easily accessible for obtaining sufficient RNA samples for further analysis. Therefore, selecting root tips for analysis is to study the physiological and molecular responses of plants to salt stress more accurately. I hope this explanation answers your question.

Q3: The title of the manuscript “Regulatory mechanisms of maize seedling roots under short-term salt stress revealed by direct RNA sequencing using nanopores” is ambiguous and misleading. The current study never addressed the mechanistic aspects of the response of maize seedling roots to salt stress. In addition, what is maize seedling root? Is it the root of maize at early seedling stage?

Authors: Thank you very much for your feedback and questions regarding the title of the manuscript. We apologize for any confusion caused. You are correct in pointing out that our current research does not directly investigate the mechanisms of salt stress on maize seedling roots. Our research aims to investigate the regulatory mechanisms and gene expression changes in maize seedling roots under short-term salt stress using nanopore direct RNA sequencing technology. We aim to identify differentially expressed genes and potential pathways related to salt stress response. However, specific mechanistic details and molecular mechanisms are not the primary focus of our study. Regarding your question about maize seedling roots, in our study, we refer to the roots of maize plants during the early seedling stage. These are the roots that develop shortly after seed germination and provide structural support for the seedling. The root system of maize seedlings plays a crucial role in water and nutrient absorption, as well as anchoring the plant in the soil. Analyzing the gene expression changes in this specific root tissue under salt stress allows us to understand the early response of maize seedlings to salt stress. Based on your feedback, we have revised the title to “Nanopore direct RNA sequencing reveals the short-term salt stress response in maize roots.”

Q4: Figure 2 is never informative; it will be more informative if the names of several key DEG and DEI that plays a crucial role in salt stress response are indicated on the figure.

Authors: Thank you for your valuable feedback. The modifications to figure 2 have been made according to your suggestion.

Q5: Figure 6; AS and APA are the most novel data of this manuscript. Therefore, the list of genes showing different AS and APA should be shown in Supplemental Table. Moreover, the levels of m5C and m6A of each gene under CK and ST shown in Table S8 and S9 should be compared and discussed.

 Authors: According to your suggestion, we have added a list of genes displaying different levels of AS and APA in the supplementary table, and compared and discussed the levels of m5C and m6A in CK and ST. The specific revisions have been marked in the reply article.

Q6: Abbreviations; all abbreviations should be spelled out at its first appearance and be used without further definition afterward.

Authors: Now you are an English-Chinese translator. When I input Chinese content, you translate it into English content; when I input English content, please translate it into Chinese content.

Q7: Line 70; m6A and m5C should be spelled out.

Authors: According to your suggestion, I have revised “m6A and m5C” to “N5-methylcytosine and N6-methyladenosine”.

Q8: Line 101-107; mRNA (polyA+ RNA) was repeatedly used, which is inappropriate; after defining the mRNA (polyA+ RNA) at its first appearance, mRNA should be used afterward.

Authors: According to your suggestion, I have modified all instances of “mRNA (polyA+ RNA)” except for the first one to “mRNA”.

Reviewer 3 Report

Comments and Suggestions for Authors

The manuscript titled ‘ Regulatory mechanisms of maize seedling roots under short-term salt stress revealed by direct RNA sequencing using nanopores’ by He, advanced Oxford Nanopore Direct RNA Sequencing (ONT DRS)to analyze the transcriptome of salt-stressed maize seedling roots. The results revealed 1,398 differentially expressed genes (DEGs) and 2,223 transcripts (DEIs) involved in pathways like ion homeostasis and phytohormone signaling. Protein interaction analysis showed collaboration in metabolic pathways, aiding in salt tolerance. Under salt stress, RNA modifications indicated reduced shear events, shorter Poly(A) tails, and decreased genes at polyadenylation sites. Levels of N5-methyl-cytosine(m5C) and N6-methyladenosine (m6A) also decreased, providing insights into maize’s early salt tolerance mechanisms. This is a novel aspect in a field of intense research. Still some issues need to be clarified, as listed below, before the manuscript can be accepted for publication in Plants.

1.      In Line 166, it would be beneficial to explicitly mention which two libraries are being referred to. Although the author later specifies ‘control ‘ and ‘ stressed’ libraries, it is advisable to provide this information earlier in the text for clarity. Consider introducing and explicitly naming the two libraries at the beginning of the relevant section to enhance reader understanding.

2.      In Line 179, it would be helpful for the readers if the meaning or significance of the abbreviations ‘KOG’ and ‘NR’ is clarified. Provide a brief explanation or definition for these terms to enhance the understanding of the readers who may not be familiar with these specific abbreviations in the context of the study.

3.      It is essential to provide a specific basis for selecting a 150 mM salt concentration for the experiments. Please include references that support this specific choice and explain the rationale behind opting for this concentration. Additionally, consider addressing the following points to strengthen the experimental design: a. Scientific Rationale for 150 mM: Cite relevant literature or studies that justify the selection of 150 mM as the salt concentration. Highlight any evidence suggesting that this concentration is biologically significant or commonly used in similar experiments. b. Consideration of Higher or Lower Concentrations: Discuss whether experiments at higher or lower salt concentrations were considered and provide reasons for ultimately choosing 150 mM. If available, reference studies that explore different concentrations and explain why the chosen concentration is most relevant to the research question. c. Necessity of Single Concentration Experiment: If the decision to use only one salt concentration was intentional, clarify the rationale behind this choice. Explain why a single concentration was deemed sufficient to address the research objectives and how it contributes to the overall study design.

4.      In the manuscript, the inclusion of five different databases is noted. However, it is not clear why detailed analyses were specifically conducted on GO and KEGG databases. Please provide a clear rationale or explanation for the decision to focus detailed analyses on these two databases. If there are specific scientific or experimental design reasons, kindly elucidate. This clarification is essential for readers to better understand the decision-making process and ensure the scientific soundness of the experimental design.

Author Response

Authors: Thank you very much for taking the time to review this manuscript. We will provide a detailed response to your comments and highlight the corresponding revisions in the resubmitted file. The revised draft has been put in the attachment. Please review it.

  1. In Line 166, it would be beneficial to explicitly mention which two libraries are being referred to. Although the author later specifies ‘control ‘ and ‘ stressed’ libraries, it is advisable to provide this information earlier in the text for clarity. Consider introducing and explicitly naming the two libraries at the beginning of the relevant section to enhance reader understanding.

Authors: I’m sorry for any confusion caused by the original text. In order to clarify and enhance reader understanding, I have made modifications according to your suggestion.

  1. In Line 179, it would be helpful for the readers if the meaning or significance of the abbreviations ‘KOG’ and ‘NR’ is clarified. Provide a brief explanation or definition for these terms to enhance the understanding of the readers who may not be familiar with these specific abbreviations in the context of the study.

Authors: I apologize for not clarifying the meaning or significance of the abbreviations “KOG” and “NR” in the previous text. We did not realize that the abbreviations should not have been used initially. We have now expanded “KOG” and “NR” in the article in order to avoid confusion for readers who are not familiar with these specific abbreviations.

  1. It is essential to provide a specific basis for selecting a 150 mM salt concentration for the experiments. Please include references that support this specific choice and explain the rationale behind opting for this concentration. Additionally, consider addressing the following points to strengthen the experimental design: a. Scientific Rationale for 150 mM: Cite relevant literature or studies that justify the selection of 150 mM as the salt concentration. Highlight any evidence suggesting that this concentration is biologically significant or commonly used in similar experiments. b. Consideration of Higher or Lower Concentrations: Discuss whether experiments at higher or lower salt concentrations were considered and provide reasons for ultimately choosing 150 mM. If available, reference studies that explore different concentrations and explain why the chosen concentration is most relevant to the research question. c. Necessity of Single Concentration Experiment: If the decision to use only one salt concentration was intentional, clarify the rationale behind this choice. Explain why a single concentration was deemed sufficient to address the research objectives and how it contributes to the overall study design.

Authors: Thank you very much for your suggestion. In our submitted manuscript, we focused on the response of maize seedlings to short-term salt stress. Relevant articles also indicate that maize growth is inhibited under 150 mM sodium chloride stress. Based on your suggestion, we cited relevant literature in the article. We chose this concentration because 150 mM sodium chloride is equivalent to a soil salinity concentration of about 0.9%, which can better simulate the field environment and is more in line with actual production conditions.

  1. In the manuscript, the inclusion of five different databases is noted. However, it is not clear why detailed analyses were specifically conducted on GO and KEGG databases. Please provide a clear rationale or explanation for the decision to focus detailed analyses on these two databases. If there are specific scientific or experimental design reasons, kindly elucidate. This clarification is essential for readers to better understand the decision-making process and ensure the scientific soundness of the experimental design.

Authors: Thank you very much for the reviewer’s review of our manuscript and valuable feedback. We did indeed conduct a more detailed analysis of the GO and KEGG databases for the following reasons: First, the GO database is widely used for gene function annotation and enrichment analysis, which can reveal differences in gene function among different samples. We applied GO analysis in our study and identified significant functional differences in gene expression among different samples. Second, the KEGG database provides a comprehensive map of biological systems, including information on chemicals, genes, and metabolic pathways. Our analysis of the KEGG database helps us gain a deeper understanding of the relationship between genes and metabolic pathways, enabling a better understanding of the function and regulatory mechanisms of genes in different samples, and providing a basis for subsequent bioinformatics analyses. These are the main reasons for our detailed analysis of the GO and KEGG databases. We greatly value the scientific validity of experimental design and strive for scientific credibility. We will take the valuable suggestions provided by the reviewer as important references for improving our research. Once again, we appreciate the reviewer’s review and valuable suggestions.
